# WHEN AND WHY VISION-LANGUAGE MODELS BEHAVE LIKE BAGS-OF-WORDS, AND WHAT TO DO ABOUT IT?

**Mert Yuksekgonul, Federico Bianchi, Pratyusha Kalluri, Dan Jurafsky, James Zou**
Stanford University
Stanford, CA 94305
`{merty, fede, pkalluri, jurafsky, jamesz}@stanford.edu`

## ABSTRACT

Despite the use of large vision and language models (VLMs) in many downstream applications, it is unclear how well they encode the compositional relationships between objects and attributes. Here, we create the Attribution, Relation, and Order (ARO) benchmark to systematically evaluate the ability of VLMs to understand different types of relationships, attributes, and order information. ARO consists of *Visual Genome Attribution*, to test the understanding of objects' properties; *Visual Genome Relation*, to test for relational understanding; and *COCO-Order & Flickr30k-Order*, to test for order sensitivity in VLMs. ARO is orders of magnitude larger than previous benchmarks of compositionality, with more than 50,000 test cases. We present the settings in which state-of-the-art VLMs behave like bags-of-words—i.e. when they have poor relational understanding, can blunder when linking objects to their attributes, and demonstrate a severe lack of order sensitivity. VLMs are predominantly trained and evaluated on large scale datasets with rich compositional structure in the images and captions. Yet, training on these datasets has not been enough to address the lack of compositional understanding, and evaluating on these datasets has failed to surface this deficiency. To understand why these limitations emerge and are not represented in the standard tests, we zoom into the training and evaluation procedures. We demonstrate that it is possible to perform well on image-text retrieval over existing datasets without using the composition and order information. This further motivates the value of using ARO to benchmark VLMs. Given that contrastive pretraining optimizes for retrieval on large datasets with similar shortcuts, we hypothesize that this can explain why the models do not need to learn to represent compositional information. This finding suggests a natural solution: composition-aware hard negative mining. We show that a simple-to-implement modification of contrastive learning significantly improves the performance on tasks requiring an understanding of order and compositionality.

## 1 INTRODUCTION

Vision and language models (VLMs) have demonstrated high performance on dozens of well-established benchmarks (Radford et al., 2021; Li et al., 2022; Singh et al., 2022; Alayrac et al., 2022; Wang et al., 2022a;b; Zhai et al., 2022). Yet it is unclear whether performance on these benchmarks indicates rich *compositional understanding* of either text or images. For example, does CLIP distinguish between "the horse is eating the grass" and "the grass is eating the horse"? Natural scenes are complex, composed of many objects and attributes, in relationships with one another. While there have been important efforts to test compositional representations of objects, attributes, and relations (Thrush et al., 2022), such efforts are based on small sets of hand-crafted examples, often combined with testing many other types of knowledge. This makes it hard to evaluate the role of relational and attributional knowledge in isolation and lacks the statistical power to quantify how well VLMs perform on granular subtypes of compositions. Here, we provide a large-scale test bed to evaluate VLMs' attribution, relation, and order understanding. Using the test bed we create, we find significant deficiencies: many models fail to perform beyond chance level at simple tasks requiring compositional understanding.

Many VLMs are pretrained and tested on large datasets with complex scenes and detailed captions with rich compositional structure. Yet, training on these datasets has not been enough to address the lack of compositional understanding, and evaluating on these datasets has failed to surface this deficiency. In the recent literature, the dominant VLM training paradigm is image-text contrastive pretraining (Jia et al., 2021; Radford et al., 2021; Zhang et al., 2020) over these large pretraining datasets. Contrastive pretraining optimizes for the task of image-text retrieval, and naturally many VLMs are tested in the retrieval task. In this work, we provide an analysis of retrieval, as an evaluation and objective. We propose experiments to analyze how these models are evaluated and trained, to understand the underlying issues.[1]

Our main contributions are three-fold:

1. **Introducing the Attribution, Relation, and Order benchmark (ARO) for fine-grained evaluation of VLMs' relation, attribution, and order understanding.** We present four new tasks: *Visual Genome Attributions* and *Visual Genome Relations*, to test the understanding of objects' attributes and relations in complex natural scenes; and *COCO Order* and *Flickr30k Order*, to test the models' ability to identify the correct ordering of the words in a caption (Section 2). Using these evaluations, we show that state-of-the-art VLMs fail to represent simple relations such as "to the right of" and "behind", fail to represent the attributive difference between "the black jacket and the blue sky" versus "the blue jacket and the black sky", and fail to represent the difference between correct and permuted captions. We provide fine-grained insights into the types of attributions and relations that models most frequently fail to understand.

2. **A critique of retrieval and contrastive pretraining.** Given we find VLMs exhibit poor compositional understanding, why have these issues not surfaced in many previous evaluations? Existing retrieval datasets are equipped with complex scenes and detailed descriptions as captions, typically full of rich compositional structure. Intriguingly, the models can perform well on retrieval without having a good compositional understanding. Our experiments (Section 3) show that models can achieve a high performance on retrieval even when the order and composition cues are removed from captions or images. Hence, it is natural that models with compositional deficiencies can still perform well on the standard evaluations. This suggests that *standard retrieval tasks are limited in their ability to assess compositional understanding of the model*, further motivating the need for our comprehensive ARO benchmark. Since contrastive pretraining optimizes for retrieval, our findings also show that models can perform well on contrastive pretraining without learning compositional information. Given our results, we argue that not learning the compositional information is a valid shortcut strategy (Geirhos et al., 2020), and VLMs have little incentive to learn to encode compositionality during contrastive pretraining.

3. **Composition-aware hard negatives can go a long way.** We propose a simple fix: mining of composition-aware hard negatives (Section 4). First, we introduce hard negatives consisting of the nearest neighboring images into each batch, to force models to represent fine-grained differences between very similar scenes. Second, we introduce hard negative captions into each batch, consisting of the true captions with word order perturbed, to force models to distinguish between correct and incorrect order. Finally, we show that this simple finetuning modification provides significant improvements in model understanding of attributes and relations.

## 2 ATTRIBUTION, RELATION, AND ORDER (ARO) BENCHMARK: WHEN DO MODELS BEHAVE LIKE A BAG-OF-WORDS?

Whereas humans effortlessly parse natural scenes containing rich objects in relation to one another, it is unclear whether machines understand the complexity of these scenes. To do so, models must be able to correctly represent objects, their attributes, and the relations between objects. Recent research has started to probe VLMs' for such information. Thrush et al. (2022) proposed Winoground, a dataset of test cases documenting a clear lack of compositional and pragmatic understanding in VLMs. The dataset is high quality but relatively small scale; its 400 test cases cover a wide range of linguistic phenomena (e.g., relation, pragmatics, world knowledge), making it hard to render statistically significant results about fine-grained relational and attributive abilities. In concurrent work, Diwan et al. (2022) suggest that Winoground has further challenges beyond compositionality

---

[1]Code is available at `github.com/mertyg/vision-language-models-are-bows`.

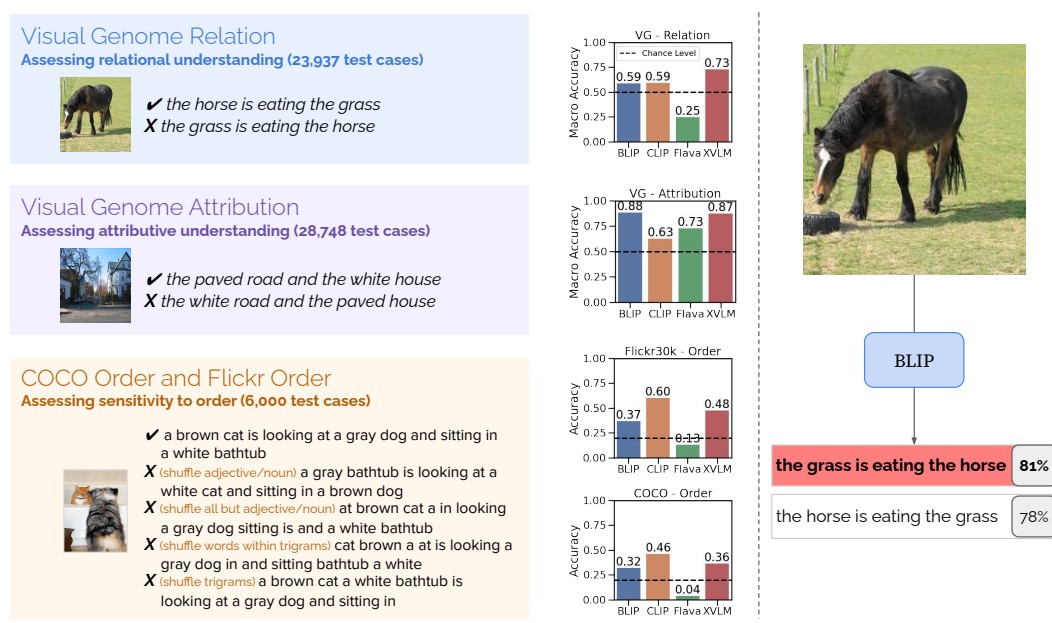

Figure 1: **ARO (Attribution, Relation and Order) a benchmark to test composition and order understanding.** We present four large-scale tasks to test the model's relational, attributive, and order understanding. These datasets probe the models' ability to pick the correct ordering of the constituents of a caption, e.g. by asking the model to pick between 'the horse is eating the grass' vs 'the grass is eating the horse'. Existing VLMs exhibit intriguing deficiencies at these simple tasks: several models remain at or below chance level. For example, BLIP chooses 'the grass is eating the horse', with 81% probability.

such as requiring common-sense reasoning or strong world knowledge. This makes it harder to render specific conclusions about compositional understanding. Here, we introduce a large dataset for a fine-grained and targeted evaluation of VLMs' ability to encode relations, attributes, and order.

## 2.1 NEW BENCHMARKS FOR ASSESSING RELATIONAL AND ATTRIBUTIVE UNDERSTANDING

We leverage Visual Genome (VG) (Krishna et al., 2017) – a large-scale dataset with over 100,000 images, annotated with objects, attributes, and relations – and the high-quality GQA annotations (Hudson & Manning, 2019) established in past work. Building upon these, we generate two novel datasets for probing relation and attribution understanding:

- **Visual Genome Relation.** Given an image and a constituent relation of the form *X relation Y*, we test whether the model can pick the correct order. Specifically, we probe models to pick between *X relation Y* and *Y relation X* with test cases from various relations, such as prepositional relations (e.g. "the **dog** is behind the **tree**' vs "the **tree** is behind the **dog**") and verbs (e.g. "the **horse** is eating the **grass**" vs "the **grass** is eating the **horse**)").

- **Visual Genome Attribution.** We test the ability to attribute properties to objects appropriately. For instance, we probe the model to pick between "the **crouched cat** and the **open door**" and "the **open cat** and the **crouched door**".

We extract images with 48 relations including *sitting on, eating, inside*, and *below*, with 23,937 test cases in total; and 117 unique attribute pairs including *'gray vs wood', 'open vs white'*, and *'small vs brown'*, with 28,748 test cases in total. The details of the dataset generation procedure are presented in Appendix A, along with the full list of relations with the count statistics. In brief, we mined Visual Genome for cases in which both of the constituent objects of the relations/attributes took a meaningfully large space in the image, and extracted the smallest bounding box containing both of the constituents, presenting this cropped image alongside the correct and swapped relation/attribute annotation. Each test case is thus made of an image (e.g., the image of a horse eating the grass) a correct caption (e.g., "the horse is eating the grass") and a grammatically correct, but swapped,

caption (e.g., "the grass is eating the horse"). For each of the test cases in these datasets, we quantify the performance of each model in identifying the correct caption from the two choices; chance level performance is 50%. Examples of these tests can be seen in Figure 1.

## 2.2 NEW BENCHMARKS FOR ASSESSING ORDER SENSITIVITY

Whereas Visual Genome Relation and Visual Genome Attribution assess the model's ability to understand order and compositionality *related to attributes and relations*, we also wish to discern whether this is connected to a broader inability to represent word order *in general*. With this motivation, we specifically want to test the order sensitivity of models. Do models broadly exhibit any preference towards the correct ordering of the words in a scene description, or are they indifferent towards any permutation, even for those that are unreasonable?

We propose an additional stress test, to test the models' ability to pick the correct ordering of the words within a caption. Given an image, we probe the models to pick between the correct ordering of a caption versus alternatives where the words are reordered in systematic ways. Given four systematic permutations of a given caption and the caption itself, can the model 'pick the right caption'? We augment existing retrieval datasets to derive *COCO Order*, and *Flickr30k Order* (Lin et al., 2014; Young et al., 2014). To generate these datasets, we utilize four different perturbations of a caption, provided in Table 1. These largely follow prior work that evaluates language models (O'Connor & Andreas, 2021). We use Spacy (Honnibal & Montani, 2017) for part-of-speech tagging to perform the perturbations.

## 2.3 EVALUATING VLMS ON ARO

We evaluate four state-of-the-art VLMs: CLIP (Radford et al., 2021), BLIP (Li et al., 2022), Flava (Singh et al., 2022), and X-VLM (Zeng et al., 2022b). More details on these models can be found in Appendix B.

**Models exhibit deficiencies in compositional understanding:** In Figure 1, we present model performance on the Visual Genome Relation and Visual Genome Attribution evaluations. In relation tests, we observe that most models are near or below chance level, indicating severe deficiencies in relational understanding. In Appendix Table 2, we provide the performance on each relation separately. For instance, while BLIP is relatively accurate at understanding positional relations, its performance is generally near chance level for verbs, such as 'eating' or 'watching'; whereas CLIP generally performs at chance level on positional relations. Quantitatively, while BLIP obtains 66% macro accuracy for spatial relations, it obtains 56% accuracy in verbs. In contrast, CLIP achieves 56% in spatial relations and 61% in verbs. In attribution tests, although BLIP (88%) and XLVM (87%) perform remarkably well, CLIP (62%) is again close to chance level. While Flava is reasonably good at attribution (73%), its performance is below-chance for relations (25%). Overall, VLMs exhibit significant deficiencies in compositional understanding, particularly relational understanding. These deficiencies motivate our interest in probing whether these models are failing to represent compositional information *in particular* — e.g. failing to represent the order of relation constituents — or whether models are in fact failing to represent word order more broadly.

Table 1: List of perturbations used in order sensitivity experiments.

| Perturbation Type | Example |
|---|---|
| Original Caption | remarkable scene with a blue ball behind a green chair |
| Shuffle nouns and adjectives | green ball with a remarkable chair behind a blue scene |
| Shuffle everything but nouns and adjectives | remarkable scene behind a blue ball with a green chair |
| Shuffle trigrams | a green chair remarkable scene with a blue ball behind |
| Shuffle words within each trigram | scene with remarkable a ball blue a green behind chair |

**Models have little to no preference toward correctly formed sentences:** In Figure 1, we present the COCO/Flickr30k order task and the performance of the tested VLMs. Given an image, the VLM must pick between the original caption and four alternative captions, augmented with the 4 perturbations listed in Table 1 respectively; thus, chance-level performance is 20%. Given the randomness over the permutations of captions, we repeat the experiment with 5 different seeds and

present the mean performance and error bars. All numerical values can be found in Appendix 5. Overall, models exhibit different levels of preference toward the correct ordering. For instance, while BLIP performs relatively well on the earlier tasks compared to CLIP, we observe here that its performance is much closer to chance level. Similarly, while Flava obtains a good performance with the Attribution task, its performance is below chance level for the COCO and Flickr30k Order task.

**Connection to prior evaluations with text-condition image generation:** We observe that CLIP cannot identify the correct ordering of constituents in a relation. This is in line with prior observations showing that text-conditioned image generators that use CLIP as the text encoder struggle with generating images that are faithful to the relations in the descriptions (Conwell & Ullman, 2022). We hypothesize the problem may lie with CLIP's inability to encode order. Given that Imagen (Saharia et al., 2022) has better results on compositionality tests, we speculate that this can be because they use T5 (Raffel et al., 2020), a language model, as the text encoder. We believe our results suggest there is potential for language model priors in VLMs to contribute increased compositional understanding.

## 3  WHY DO MODELS BEHAVE LIKE BAG-OF-WORDS? A CRITIQUE OF RETRIEVAL AND CONTRASTIVE PRETRAINING

Given that VLMs exhibit poor compositional and order understanding, why have these issues not surfaced in many previous evaluations? Most VLMs are consistently evaluated on image-to-text retrieval, and demonstrate high performance. In this section, we demonstrate why retrieval can be incomplete, both as an evaluation and as an objective. We first show that models can perform well in the text-image retrieval evaluations on existing large-scale datasets, without using order or composition information. It is thus natural that the issues of lack of compositional and order information have been masked by the high performance. Next, we discuss the connection between contrastive pretraining on large datasets and the task of retrieval and argue that models may not have an incentive to learn composition and order.

### 3.1  LIMITATIONS OF RETRIEVAL AS AN EVALUATION

Even though existing retrieval datasets are equipped with complex scenes and detailed descriptions, it is unclear how much complexity the models need to understand to perform well on this task. In particular, do the models have to use compositional information to perform well on a large-scale retrieval task? To understand what it does and does not take to perform well on retrieval, we propose evaluations with modified datasets. First, we propose augmentations to the existing datasets, where we remove the order and composition cues. Next, we evaluate models on these augmented versions of these datasets to understand whether it is possible to perform well without these cues.

**Datasets:** We zoom into two of the standard cross-modal text-image retrieval datasets, namely COCO (Lin et al., 2014) and Flickr30k (Young et al., 2014). Following prior work (Li et al., 2022), we use Karpathy splits (Karpathy & Fei-Fei, 2015) for both of the datasets. We test the models on the test splits; where COCO contains 5k images, and Flickr30k contains 1k images. We report Recall@1 and Recall@5 for both datasets.

**Experimental Protocol:** Our goal is to understand whether models need order information / compositional understanding to perform well on existing text-image retrieval datasets. We test our hypothesis on the augmented versions of the existing datasets. We propose two augmented setups (see Figure 2):

1. **Perturbing the order and composition information in the captions:** To understand if models need the compositional information in captions to perform well, we aim to remove these, and test whether models can perform well without them. In this case, we take the permutations of the words in a given caption, using the strategies specified in Table 1. For instance, we take the COCO dataset, shuffle all of the words in all of the captions, and compute the retrieval performance over this caption-modified dataset. Given that all words are shuffled, the compositional structure within the captions is altered, e.g. "the grass is eating the horse" becomes "eating the grass horse the".

2. **Perturbing the order and composition information in the images:** Similar to captions, to understand if models need the compositional information in the images to perform well, we aim to test the models in the absence of these. As it is harder to manipulate entities in an image, we resort to a more severe method and take the permutations over the patches of an image. For

instance, we split images into 9 equally sized patches and take a permutation of those patches to form the new images. Qin et al. (2021) used a similar strategy to show the insensitivity of vision transformers to patch-based augmentations, in the context of image classification. Again, this should alter most compositional structures within an image; for instance, an object that is "below" another can move into an arbitrary point in an image. We then compute the retrieval performance with this image-modified dataset. We explore three such strategies: splitting the image into either 4 equally-sized rows, 4 equally-sized columns, or 9 equally-sized patches and then shuffling.

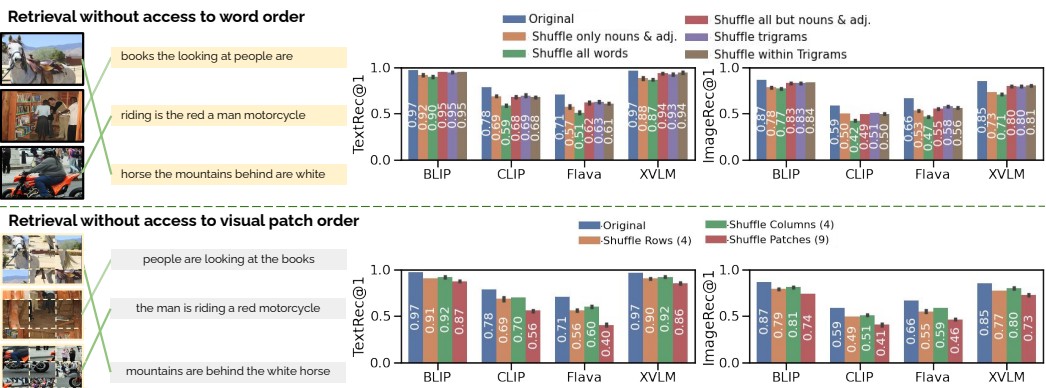

Figure 2: **Retrieval without access to order information.** We show that models can achieve substantially high performance on standard evaluations even when order information is removed. In particular, in datasets where the captions are augmented with order perturbations, models show marginal performance degradation.

**Models can achieve high performance even when order information is inaccessible.** In Figure 2, we present the retrieval performance of existing models under different augmentation strategies, and a detailed table can be found in Appendix 3,4. For each perturbation, we augment the dataset and compute the retrieval performance 3 times with different random seeds, and report standard error bars. Notably, across all of the mentioned perturbation strategies, most models lose marginal performance in performing the retrieval task with the perturbed caption, or the perturbed image. These results demonstrate that it is possible to obtain high performance in a retrieval task without utilizing the compositional structure, even in the case of these arguably large-scale datasets.

## 3.2 LIMITATIONS OF RETRIEVAL AND CONTRASTIVE PRETRAINING AS AN OBJECTIVE

To understand why these deficiencies emerge in the first place, we discuss the training procedure of VLMs. Most state-of-the-art VLMs, at their core, are trained with a contrastive loss (Radford et al., 2021; Chen et al., 2020; Zeng et al., 2022a; Li et al., 2021; 2022; Zhang et al., 2020) on a large pretraining dataset scraped from the web. We hypothesize that the lack of compositional understanding can be attributed to the way the models are trained. Here, we discuss the connection between contrastive pretraining to retrieval, to better understand the underlying phenomenon.

Quoting from Radford et al. (2021): "CLIP pre-trains for the task of image-text retrieval on our noisy web-scale dataset". The goal in contrastive text-image pretraining is to optimize the model to identify the matching pairs of caption and text; namely *retrieval*. Simple as it is, training models with a retrieval objective, on large pretraining datasets, has demonstrated remarkable off-the-shelf performance, for instance in tasks requiring single object recognition.

Although the existing large retrieval/pretraining datasets can have long and detailed captions, our results demonstrate that it is possible to perform well on these datasets without using compositional structure. Accordingly, these results provide evidence that *models can achieve high performance in retrieval objectives, thus also obtaining low contrastive loss, without order information, unless the datasets are carefully designed.* These datasets are designed to cover a large conceptual and semantic space, in order to make models broadly useful for downstream tasks; yet these datasets are not designed to contain many images with captions containing similar words that must be differentiated. Without such alternatives in the dataset, the task can be solved without taking order information into account — and behaving like a *bag-of-words* becomes a high-reward strategy. There is a large body of evidence showing that neural networks are prone to exploiting shortcut strategies (Geirhos et al.,

2020; 2019), and our findings demonstrate that not learning the order information is a valid shortcut strategy, for general-purpose retrieval/captioning datasets. *We thus argue that it is unclear what should incentivize models trained with a contrastive loss to learn to pay attention to order structure unless the datasets or algorithms are carefully designed with this consideration.*

## 4 A SIMPLE FIX: COMPOSITION-AWARE HARD NEGATIVES

Our analysis of the retrieval objective and datasets leads to a natural solution: hard negatives for contrastive learning (Robinson et al., 2021; Kalantidis et al., 2020). We propose a natural extension of CLIP's contrastive objective to alleviate some of the issues discovered in previous sections. To make CLIP sensitive to word order and better at capturing compositions, we use strong alternatives:

1. **Generation of negative captions**: For each image-caption pair, we generate a negative caption by swapping different linguistic elements: *noun phrases, nouns, adjectives, adverbs, verb phrases*. For example, the caption "The horse is eating the grass and the zebra is drinking the water" either becomes "The zebra is eating the grass and the horse is drinking the water" (noun swapping) or "The horse is drinking the grass and the zebra is eating the water" (verb phrase swapping).

2. **Sampling strong alternative images:** To generate images that are strong alternatives to the images in a batch, we first use CLIP to compute the pairwise similarity between all images in the dataset. During training, for each image in the batch, we sample one of the $K = 3$ nearest neighbors as the strong alternative image. The sampled alternative images (and the respective captions and negative captions) are added to the batch.

We first briefly describe the standard contrastive methodology used in CLIP: Let $f_i : \mathcal{X}_{\text{image}} \to \mathbb{R}^d$ be the image encoder and $f_t : \mathcal{X}_{\text{text}} \to \mathbb{R}^d$ be the text encoder for a VLM. In the CLIP objective, given a batch of $N$ image-caption pairs, the goal is to predict the true pairings. Given a batch of $N$ images $\mathbf{I}_N = \{I_1, I_2, ..., I_N\}$ and $N$ captions $\mathbf{T}_N = \{T_1, T_2, ..., T_N\}$, CLIP first computes the matrix of cosine similarities, denoted by $\mathbf{S} \in \mathbb{R}^{N \times N}$, where each item $S_{j,k}$ computes the cosine similarity between the image $j$ and caption $k$. Using these similarities, the row-wise and column-wise cross-entropy losses are computed.

For composition-aware mining of hard negatives, we propose a simple modification to the CLIP objective. In Figure 3 we give an overview of the procedure. First, given a batch of images $\mathbf{I}_N$ and captions $\mathbf{T}_N$, we generate negative captions $\mathbf{T}_N^-$, and then concatenate the two sets to obtain $\tilde{\mathbf{T}}_{2N}$. Next, we again compute the similarity matrix $\tilde{\mathbf{S}} \in \mathbb{R}^{N \times 2N}$. Here, the row-wise and column-wise cross-entropy losses are computed as in CLIP, with the difference that we do not compute the loss for the negative captions column-wise (as there is no matching image for a negative caption).

**Experimental protocol:** Due to the computational cost of training CLIP from scratch, we focused on finetuning experiments. Specifically, we finetune the ViT-B/32 variant of CLIP on the COCO dataset with hard negatives (NegCLIP). As an ablation, we also perform finetuning on COCO, without the sampled hard negatives to disentangle the effect of finetuning. The details of finetuning, hyperparameter selection, and ablation are provided in Appendix C.

**Evaluation:** We propose two main sets of evaluations. First, we evaluate models on the four order and composition-sensitive tasks, namely *Visual Genome Relation*, *Visual Genome Attribution*, *COCO & Flickr30k Order*. In addition to these, to ensure that the model is still comparable to the original CLIP, we perform evaluations on five downstream tasks: CIFAR10, 100 (Krizhevsky et al., 2009) and ImageNet (Deng et al., 2009) for image classification; and Flickr30k and COCO for retrieval.

**Results:** In Figure 3, we provide a comparison of CLIP to NegCLIP with a radar plot for an overview; numerical values can be found in Appendix Table 6 with the additional ablation on a CLIP model fine-tuned on MSCOCO without negative samples. NegCLIP does not suffer in downstream tasks, and it improves the performance on VG-Relation from 63% to 81%, on VG-Attribution from 62% to 71%, on COCO Order from 46% to 86%, and on Flickr30k Order from 59% to 91%. Further, in VG-Relations, NegCLIP becomes the best model, in comparison to all other models, and in VG-Attribution it becomes comparable to X-VLM and BLIP.

Overall, we observe that NegCLIP does not have a substantial loss in performance on the downstream tasks, yet provides substantial gains on the order-sensitive tasks. Our results highlight that

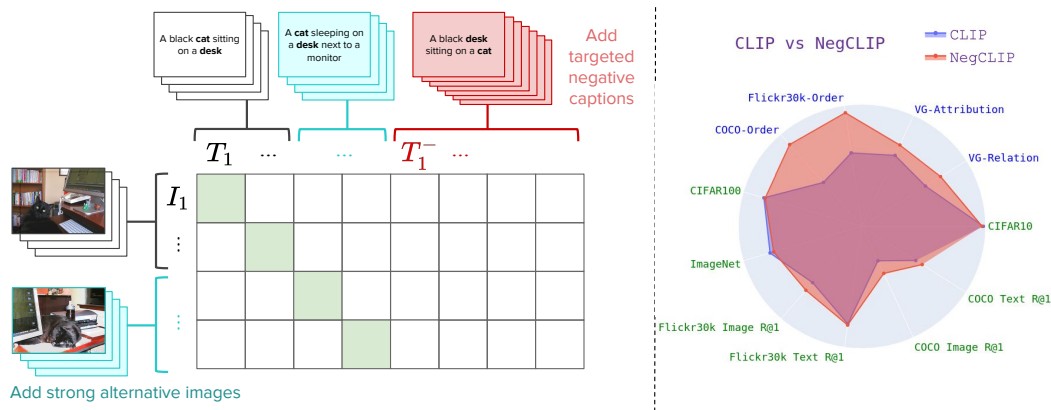

Figure 3: **Finetuning CLIP with targeted alternatives.** We propose a straightforward extension of CLIP. For each image, we sample strong alternatives among the dataset using nearest neighbors, and we create targeted negative captions to enhance order sensitivity. This method improves CLIP in a suite of compositional tasks, while not substantially hurting performance in important downstream tasks. Text in blue: tasks we proposed. Text in green: standard downstream evaluations.

mining targeted and cheap negatives can lead to substantial improvements in compositional tasks, without causing a loss of performance in existing downstream tasks. While contrastive learning provided substantial increases in representation learning, it is unclear if simply scaling the size of pretraining datasets will be as efficient as exploring algorithmic improvements, especially in learning compositional structure. We believe our results provide evidence that seeking such modifications can bring various increases in model capability. We do not suggest that the model presented here is the best possible model for encoding relations; rather, when the goal is to represent these bits of compositional information, our addition of targeted negatives should be viewed as a significant avenue for improving contrastive learning.

## 5 RELATED WORK

**Visio-linguistic compositionality:** Understanding what aspects of language and vision VLMs capture is the main objective of several recent papers. Frank et al. (2021) suggest that information sharing between the text and vision modalities is not balanced, as the representations from the text encoder are more influenced by the vision modality than the other way round. Parcalabescu et al. (2021b) show that VLMs have difficulties in counting objects in images. In terms of the evaluation part of our paper, Winoground (Thrush et al., 2022) presents the nearest neighbor to our work. Winoground is a carefully curated dataset of 400 test cases that aims to evaluate compositional and pragmatics language understanding of VLMs where VLMs perform around chance level over a set of image-text matching tasks. Diwan et al. (2022) suggest that Winoground has challenges beyond compositionality that requires complex reasoning with visually/textually difficult or unusual examples. Our main goal is to propose an isolated fine-grained evaluation of relation and attribution understanding with a large scale dataset; hence we believe our dataset complements Winoground. While Winoground proposes a smaller scale dataset with great breadth across types of model knowledge; we attempt to focus on a large scale dataset for in-depth testing for relation and attributive understanding.

Bogin et al. (2021) utilizes VG to test VQA systems for compositional generalization; in comparison here our purpose is to implement tasks that can probe any VLM. Closer in spirit to our VG - Relations dataset, Conwell & Ullman (2022) demonstrate the lack of competency of a text-to-image model, DALL-E, in generating images faithful to the relationships described in the textual prompt. Zerroug et al. (2022) design a benchmark to evaluate vision models' understanding of simple compositional attributes of abstract shapes. GQA (Hudson & Manning, 2019) is a visual question-answering dataset derived from Visual Genome (Krishna et al., 2017) to test visual question answering systems for scene understanding with compositional question answering. Other VQA datasets such as NVLR (Suhr et al., 2017; 2018) cover a broad range of linguistic phenoma including compositionality. VALSE (Parcalabescu et al., 2021a) is a dataset that tests whether VLMs can identify the correct

linguistic phenomenon appearing in an image. Recently, Saharia et al. (2022) released a set of prompts, collectively called DrawBench, as a benchmark for text-to-image models' ability to generate images faithful to the given challenging prompts, involving a set of compositional tests. Previous evaluations, while informative, were limited by small data size; Winoground has 400 examples, limiting the statistical strength of the fine-grained conclusions. Our ARO dataset (50,000 test cases) is two orders of magnitude larger, enabling us to characterize the performance for different types of compositions. Moreover, previous works did not quantify how using composition-aware hard negatives in contrastive learning could affect VLMs' performance for compositional tasks.

**Order information in language and vision:** Several works have highlighted the lack of word order sensitivity in large language models (Hessel & Schofield, 2021; O'Connor & Andreas, 2021; Pham et al., 2021; Sinha et al., 2021). Sinha et al. (2021) show that pre-training BERT with sentences with shuffled words, marginally affect the performance on down stream tasks. Pham et al. (2021) show that some of the tasks in GLUE (Wang et al., 2018) can be solved even when disregarding word order. Our analysis of image retrieval leans in a very similar direction, further suggesting the need for more careful benchmarks. O'Connor & Andreas (2021) show that for long-range contexts, models use content words and local co-occurrence statistics to make predictions. Ettinger (2020) uses a set of psycholinguistic tasks to evaluate the linguistic and contextual information used by BERT, showing BERT's insensitivity to elements like negation. On the vision side, Brendel & Bethge (2019) show that a bag-of-local-features model performs almost as well as their state-of-the-art counterparts. Closer to our experiments, the work of Tejankar et al. (2021) shows that training contrastive vision language models using only bag-of-words in place of the caption does not significantly hurt performance on zero-shot classification. Our work generalize these results, showcasing the general limits of vision language models when dealing with relations, attributes and shuffled captions.

**Negative mining and contrastive learning**: Using hard negatives has been successful in improving representation learning (Harwood et al., 2017; Wu et al., 2017; Ge, 2018). Furthermore, hard negatives have also been shown to improve contrastive learning (Kalantidis et al., 2020; Robinson et al., 2021) or used with a contrastive loss to improve ViTs (Qin et al., 2021). Our work differs in that we propose exploring hard negatives with contrastive learning, particularly in the context of vision-language models and compositional abilities. Note that Li et al. (2021) and Li et al. (2022) use negative mining by selecting pairs of items with high similarity. However, in light of our results, this strategy alone does not seem to be enough to effectively train the model to deal with relationships and word order.

# 6    CONCLUSION

In this work, we evaluate the ability of VLMs to encode composition and order structure, introducing large-scale test beds to generate fine-grained and statistically strong insights. We show that models struggle with relation, attribution, and order understanding, and our datasets revealed various limitations of models. We show that models can achieve high performance on the task of cross-modal retrieval without needing to learn order and composition information. Given that contrastive pretraining optimizes models for the task of retrieval, we argue that this can explain why VLMs need not learn to encode order and compositional cues. Using these insights, we presented a simple modification to the training procedure, namely composition-aware hard negative mining. Through several evaluations, we demonstrate that by generating composition-aware hard negatives during model training, the compositional and order understanding of VLMs can be improved.

Our work demonstrates the importance of the interaction between the pretraining objective and large datasets VLMs are trained on. In this work, we focused on finetuning of VLMs for demonstrating the use of the composition-aware negative mining. For future work, we are interested in further exploring composition-aware contrastive pretraining of VLMs. Given that VLMs are trained with large datasets with rich text corpora, the limited language understanding of these models are intriguing. While here we specifically focused on contrastive learning, in light of our findings, studying the interaction between different pretraining objectives and compositional understanding is an emerging future avenue. Our results further highlight the importance of rich evaluations of VLMs. We hope future VLMs will release results on these fine-grained evaluations in addition to standard tasks, and more fine-grained evaluations will be developed. Focused evaluation of state of the art models illuminates the strengths and deficiencies of these models, and is key to understanding in which contexts and for which goals these models can be used.

## ACKNOWLEDGMENTS

We would like to thank Adarsh Jeewajee, Candace Ross, Duygu Yilmaz, Edward Chen, Kyle Swanson, Rishi Bommasani, Tristan Thrush, Tuomas Oikarinen, and Weixin Liang for their support and comments on the manuscript, and all members of the Zou Lab, Jurafsky Lab, and Guestrin Lab for helpful discussions. We thank the anonymous reviewers for their suggestions to improve the paper. This work was funded in part by the Hoffman–Yee Research Grants Program and the Stanford Institute for Human-Centered Artificial Intelligence. P.K. is supported in part by the Open Philanthropy AI Fellowship. J.Z. is supported by NSF CAREER 1942926 and the Chan-Zuckerberg Biohub.

## ETHICAL STATEMENT

There are substantial critiques of image datasets established in prior literature for lacking careful consideration of privacy and stereotypical representations of people (Peng et al., 2021; Birhane & Prabhu, 2021; Krishna et al., 2017), as well as critiques of the use of significant resources needed to train and evaluate models on such large datasets. We do not introduce any new images, so avoid introducing substantial *new* data concerns; yet, in order to facilitate comparison with prior work, the datasets in this paper are based on and use these standard, existing datasets, perpetuating their use.

More broadly, the goals and downstream consequences of papers like ours have key ethical dimensions. A central goal of this paper is to challenge broad assertions of high performance and illuminate specific strengths and deficiencies of state of the art machine learning models, which can contribute to understanding and advocating for which contexts and which goals these models should and should not be used. Understanding strengths and deficiencies of vision and language models like CLIP in particular is increasingly important as these models have become core components of text-to-image generation models, which are now being used by millions of users to generate images, for personal, creative, or commercial purposes (OpenAI, 2022). Everyday users are frequently confronted with the compositional failings of these models. Moreover, early evidence suggests when these models fail to correctly represent compositional information (like attribution of properties to entities), they may default to stereotypes; e.g., Bianchi et al. (2022) notes a case in which DallE fails to represent the compositional information in the prompt "a disabled woman leading a meting", instead generating an image attributing the property "disabled" to an audience member and the property of able-bodiedness to the meeting leader; this is in contrast to other more conventional attributes such as in "a blonde woman leading a meting" correctly generating images of blonde leaders. Highlighting technical and social failings and their uneven distribution across people can assist in advocating to reform, avoid, or reject the use of these models, and this connects to a broader body of literature directly exposing many biases and stereotypes perpetuated by these models (Cho et al., 2022; Bansal et al., 2022; Wolfe et al., 2022). Crucially, beyond contributing to the more visible and emerging experience of everyday users, vision language models serve as a modern iteration of image-classification models. Accordingly, improving model capabilities is likely to contribute to institutions' more obscured, historical and ongoing use of extracted materials and labor for the design of machine-learning-based surveillance. This is emphasized, for example, by the use of models in prior work and this work to uncritically label humans in everyday situations (Raji & Fried, 2021; Broussard, 2018). It is a complex, ongoing responsibility for machine learning researchers like ourselves to work to understand and carefully consider the possible and actual use of our evaluation of these models and the models themselves.

## REPRODUCIBILITY STATEMENT

The code to reproduce all experiments, along with the code to generate the datasets and tasks we propose are released at https://github.com/mertyg/vision-language-models-are-bows Experiments on caption perturbation have been run with three different seeds to take into account the randomness of the permutation methods. All of the models that are used were obtained from the checkpoints released with the respective

paper. Particularly, we obtained the checkpoints released at BLIP[2], X-VLM[3], CLIP[4], and Flava[5] repositories.

Experiments with CLIP and NegCLIP have not been run multiple times due to the computational requirements. However, the experiments have been run using fixed seed, so they can be replicated by other researchers. In addition to this, our NegCLIP is implemented as a fork of the open clip project[6]. The code will be released with the same license.

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

# A  PROPOSED DATASETS

## A.1  GENERATING THE DATASETS

We follow the below steps while generating the datasets:

1. We go through the scene graphs annotated in GQA (Hudson & Manning, 2019).

2. First, we identify candidate objects in all scenes. Namely, we require the objects to be large enough such that they are recognizable in an image. We enforce a heuristical criterion, where we discard all objects that have a width lower than $\frac{1}{4}$ of the width of the entire image, or with a height lower than $\frac{1}{4}$ of the height of the entire image.

3. For VG-Relation, we identify all pairings of objects, where we make sure that the pair of objects are not from the same category (say, both are not simultaneously "dogs"). Similarly, for VG-Attribution, we identify all pairs of objects that are modified by at least 1 attribute, where both objects and attributes are different from each other.

4. After identifying the pairs of objects, we extract the smallest bounding box containing both of the objects from the scene. This is to minimize the distraction in the rich scenes in Visual Genome.

5. Finally, for each identified pairs of objects, we fill the preset templates. For relations, we fill the templates of "the [object 1] is [relation] [object 2]" for the true caption, and "the [object 2] is [relation] [object 1]" for the false caption. For attributes, we fill the templates of "the [attribute 1] [object 1] and the [attribute 2] [object 2]" and "the [attribute 2] [object 1] and the [attribute 1] [object 2]".

6. For Visual Genome Relations, we post-process the relations to remove symmetric relations; such as "near" or "next to".

Overall, this process results in a set of $23,937$ test cases for relations, and $28,748$ test cases for attributes.

Table 2: Fine-grained results in Visual Genome Relation dataset.

| | CLIP | NegCLIP | CLIP-FT | XVLM | BLIP | Flava | # Samples |
|---|---|---|---|---|---|---|---|
| **Accuracy** | 0.59 | 0.8 | 0.64 | 0.73 | 0.59 | 0.24 | |
| Spatial Relationships | | | | | | | |
| **Accuracy** | 0.56 | 0.66 | 0.57 | 0.74 | 0.66 | 0.34 | |
| above | 0.48 | 0.60 | 0.54 | 0.80 | 0.64 | 0.55 | 269 |
| at | 0.59 | 0.93 | 0.71 | 0.72 | 0.49 | 0.15 | 75 |
| behind | 0.56 | 0.29 | 0.34 | 0.82 | 0.77 | 0.28 | 574 |
| below | 0.56 | 0.46 | 0.48 | 0.74 | 0.69 | 0.44 | 209 |
| beneath | 0.80 | 0.70 | 0.70 | 0.80 | 0.70 | 0.40 | 10 |
| in | 0.63 | 0.89 | 0.63 | 0.73 | 0.72 | 0.09 | 708 |
| in front of | 0.54 | 0.75 | 0.70 | 0.66 | 0.55 | 0.78 | 588 |
| inside | 0.50 | 0.91 | 0.67 | 0.69 | 0.72 | 0.12 | 58 |
| on | 0.52 | 0.86 | 0.58 | 0.86 | 0.76 | 0.12 | 1684 |
| on top of | 0.43 | 0.75 | 0.58 | 0.85 | 0.79 | 0.19 | 201 |
| to the left of | 0.49 | 0.50 | 0.50 | 0.52 | 0.51 | 0.50 | 7741 |
| to the right of | 0.49 | 0.50 | 0.50 | 0.52 | 0.49 | 0.51 | 7741 |
| under | 0.64 | 0.43 | 0.54 | 0.86 | 0.73 | 0.27 | 132 |
| Verbs | | | | | | | |
| **Accuracy** | 0.61 | 0.86 | 0.66 | 0.73 | 0.56 | 0.2 | |
| carrying | 0.33 | 0.83 | 0.75 | 0.75 | 0.67 | 0.08 | 12 |
| covered by | 0.47 | 0.36 | 0.36 | 0.61 | 0.58 | 0.56 | 36 |
| covered in | 0.79 | 0.50 | 0.50 | 0.14 | 0.29 | 0.14 | 14 |
| covered with | 0.56 | 0.56 | 0.50 | 0.56 | 0.50 | 0.19 | 16 |
| covering | 0.39 | 0.58 | 0.45 | 0.67 | 0.55 | 0.06 | 33 |
| cutting | 0.75 | 0.83 | 0.83 | 0.67 | 0.25 | 0.00 | 12 |
| eating | 0.57 | 1.00 | 0.67 | 0.62 | 0.52 | 0.00 | 21 |
| feeding | 0.90 | 0.80 | 0.80 | 0.60 | 0.30 | 0.20 | 10 |
| grazing on | 0.10 | 0.90 | 0.30 | 0.60 | 0.40 | 0.50 | 10 |
| hanging on | 0.79 | 1.00 | 0.93 | 0.93 | 0.79 | 0.00 | 14 |
| holding | 0.58 | 0.97 | 0.79 | 0.67 | 0.44 | 0.27 | 142 |
| leaning on | 0.67 | 1.00 | 1.00 | 0.75 | 0.58 | 0.08 | 12 |
| looking at | 0.84 | 1.00 | 0.68 | 0.68 | 0.55 | 0.26 | 31 |
| lying in | 0.47 | 1.00 | 0.60 | 0.87 | 0.67 | 0.00 | 15 |
| lying on | 0.60 | 0.88 | 0.50 | 0.93 | 0.75 | 0.17 | 60 |
| parked on | 0.67 | 0.86 | 0.38 | 0.76 | 0.86 | 0.00 | 21 |
| reflected in | 0.64 | 0.71 | 0.57 | 0.50 | 0.43 | 0.43 | 14 |
| resting on | 0.38 | 0.85 | 0.23 | 0.92 | 0.54 | 0.15 | 13 |
| riding | 0.71 | 0.98 | 0.78 | 0.82 | 0.41 | 0.02 | 51 |
| sitting at | 0.62 | 1.00 | 0.88 | 0.88 | 0.46 | 0.00 | 26 |
| sitting in | 0.57 | 0.96 | 0.78 | 0.87 | 0.83 | 0.30 | 23 |
| sitting on | 0.58 | 0.97 | 0.78 | 0.94 | 0.73 | 0.14 | 175 |
| sitting on top of | 0.50 | 0.90 | 0.80 | 1.00 | 0.80 | 0.10 | 10 |
| standing by | 0.67 | 0.92 | 0.67 | 0.83 | 0.67 | 0.67 | 12 |
| standing in | 0.73 | 0.98 | 0.69 | 0.69 | 0.49 | 0.05 | 59 |
| standing on | 0.60 | 1.00 | 0.63 | 0.83 | 0.73 | 0.06 | 52 |
| surrounded by | 0.64 | 0.71 | 0.64 | 0.71 | 0.64 | 0.79 | 14 |
| using | 0.84 | 1.00 | 1.00 | 0.68 | 0.58 | 0.00 | 19 |
| walking in | 0.70 | 1.00 | 0.70 | 0.60 | 0.50 | 0.00 | 10 |
| walking on | 0.79 | 1.00 | 0.79 | 0.84 | 0.42 | 0.05 | 19 |
| watching | 0.45 | 0.55 | 0.27 | 0.59 | 0.68 | 0.36 | 22 |
| wearing | 0.47 | 0.99 | 0.88 | 0.68 | 0.48 | 0.64 | 949 |

## B MODELS

1. CLIP (Radford et al., 2021): We use the 'ViT-B/32' variant of CLIP, released at https://github.com/openai/CLIP/.

Table 3: Performance in the text shuffled retrieval task. Performance is averaged over 3 seeds, and standard deviations are reported next to each mean.

| Strategy | Model | Text Recall@1 | Text Recall@5 | Image Recall@1 | Image Recall@5 |
|---|---|---|---|---|---|
| COCO Text Shuffle | | | | | |
| No Shuffling | BLIP | 0.814 | 0.953 | 0.635 | 0.855 |
| Shuffle only nouns and adj. | BLIP | $0.710 \pm 0.006$ | $0.907 \pm 0.001$ | $0.513 \pm 0.001$ | $0.768 \pm 0.001$ |
| Shuffle all words | BLIP | $0.690 \pm 0.004$ | $0.899 \pm 0.002$ | $0.505 \pm 0.002$ | $0.763 \pm 0.001$ |
| Shuffle all but nouns and adj. | BLIP | $0.767 \pm 0.002$ | $0.935 \pm 0.001$ | $0.579 \pm 0.000$ | $0.817 \pm 0.000$ |
| Shuffle trigrams | BLIP | $0.762 \pm 0.001$ | $0.933 \pm 0.000$ | $0.581 \pm 0.000$ | $0.816 \pm 0.000$ |
| Shuffle within Trigrams | BLIP | $0.767 \pm 0.003$ | $0.934 \pm 0.000$ | $0.585 \pm 0.000$ | $0.820 \pm 0.000$ |
| No Shuffling | CLIP | 0.503 | 0.748 | 0.301 | 0.557 |
| Shuffle only nouns and adj. | CLIP | $0.420 \pm 0.003$ | $0.684 \pm 0.007$ | $0.244 \pm 0.002$ | $0.480 \pm 0.001$ |
| Shuffle all words | CLIP | $0.341 \pm 0.001$ | $0.608 \pm 0.005$ | $0.205 \pm 0.001$ | $0.422 \pm 0.003$ |
| Shuffle all but nouns and adj. | CLIP | $0.415 \pm 0.002$ | $0.671 \pm 0.001$ | $0.248 \pm 0.002$ | $0.483 \pm 0.001$ |
| Shuffle trigrams | CLIP | $0.411 \pm 0.006$ | $0.673 \pm 0.004$ | $0.251 \pm 0.001$ | $0.490 \pm 0.002$ |
| Shuffle within Trigrams | CLIP | $0.404 \pm 0.002$ | $0.661 \pm 0.004$ | $0.243 \pm 0.001$ | $0.478 \pm 0.001$ |
| No Shuffling | Flava | 0.454 | 0.788 | 0.388 | 0.682 |
| Shuffle only nouns and adj. | Flava | $0.335 \pm 0.003$ | $0.645 \pm 0.002$ | $0.287 \pm 0.001$ | $0.566 \pm 0.000$ |
| Shuffle all words | Flava | $0.338 \pm 0.006$ | $0.631 \pm 0.001$ | $0.260 \pm 0.001$ | $0.526 \pm 0.002$ |
| Shuffle all but nouns and adj. | Flava | $0.392 \pm 0.005$ | $0.692 \pm 0.003$ | $0.317 \pm 0.002$ | $0.601 \pm 0.001$ |
| Shuffle trigrams | Flava | $0.385 \pm 0.006$ | $0.698 \pm 0.007$ | $0.333 \pm 0.002$ | $0.621 \pm 0.001$ |
| Shuffle within Trigrams | Flava | $0.394 \pm 0.005$ | $0.698 \pm 0.000$ | $0.324 \pm 0.001$ | $0.606 \pm 0.001$ |
| No Shuffling | XVLM | 0.791 | 0.947 | 0.610 | 0.848 |
| Shuffle only nouns and adj. | XVLM | $0.655 \pm 0.005$ | $0.884 \pm 0.000$ | $0.462 \pm 0.001$ | $0.731 \pm 0.000$ |
| Shuffle all words | XVLM | $0.633 \pm 0.006$ | $0.879 \pm 0.001$ | $0.450 \pm 0.002$ | $0.723 \pm 0.002$ |
| Shuffle all but nouns and adj. | XVLM | $0.734 \pm 0.004$ | $0.928 \pm 0.002$ | $0.547 \pm 0.003$ | $0.803 \pm 0.000$ |
| Shuffle trigrams | XVLM | $0.727 \pm 0.004$ | $0.920 \pm 0.002$ | $0.544 \pm 0.001$ | $0.799 \pm 0.004$ |
| Shuffle within Trigrams | XVLM | $0.739 \pm 0.007$ | $0.929 \pm 0.004$ | $0.554 \pm 0.005$ | $0.808 \pm 0.001$ |
| Flickr30k Text Shuffle | | | | | |
| No Shuffling | BLIP | 0.972 | 0.999 | 0.869 | 0.974 |
| Shuffle only nouns and adj. | BLIP | $0.919 \pm 0.008$ | $0.993 \pm 0.004$ | $0.786 \pm 0.003$ | $0.949 \pm 0.001$ |
| Shuffle all words | BLIP | $0.902 \pm 0.008$ | $0.988 \pm 0.003$ | $0.770 \pm 0.002$ | $0.939 \pm 0.001$ |
| Shuffle all but nouns and adj. | BLIP | $0.950 \pm 0.003$ | $0.996 \pm 0.002$ | $0.829 \pm 0.006$ | $0.961 \pm 0.001$ |
| Shuffle trigrams | BLIP | $0.948 \pm 0.005$ | $0.997 \pm 0.001$ | $0.828 \pm 0.001$ | $0.965 \pm 0.002$ |
| Shuffle within Trigrams | BLIP | $0.953 \pm 0.004$ | $0.997 \pm 0.001$ | $0.838 \pm 0.002$ | $0.964 \pm 0.001$ |
| No Shuffling | CLIP | 0.784 | 0.950 | 0.591 | 0.835 |
| Shuffle only nouns and adj. | CLIP | $0.690 \pm 0.005$ | $0.909 \pm 0.001$ | $0.501 \pm 0.003$ | $0.770 \pm 0.003$ |
| Shuffle all words | CLIP | $0.587 \pm 0.007$ | $0.854 \pm 0.011$ | $0.423 \pm 0.006$ | $0.694 \pm 0.002$ |
| Shuffle all but nouns and adj. | CLIP | $0.678 \pm 0.010$ | $0.904 \pm 0.007$ | $0.493 \pm 0.003$ | $0.764 \pm 0.002$ |
| Shuffle trigrams | CLIP | $0.698 \pm 0.017$ | $0.910 \pm 0.005$ | $0.509 \pm 0.004$ | $0.775 \pm 0.002$ |
| Shuffle within Trigrams | CLIP | $0.680 \pm 0.006$ | $0.903 \pm 0.011$ | $0.498 \pm 0.006$ | $0.766 \pm 0.001$ |
| No Shuffling | Flava | 0.707 | 0.941 | 0.664 | 0.900 |
| Shuffle only nouns and adj. | Flava | $0.573 \pm 0.021$ | $0.869 \pm 0.009$ | $0.532 \pm 0.001$ | $0.816 \pm 0.002$ |
| Shuffle all words | Flava | $0.504 \pm 0.002$ | $0.817 \pm 0.009$ | $0.467 \pm 0.006$ | $0.754 \pm 0.006$ |
| Shuffle all but nouns and adj. | Flava | $0.622 \pm 0.016$ | $0.888 \pm 0.005$ | $0.553 \pm 0.004$ | $0.823 \pm 0.002$ |
| Shuffle trigrams | Flava | $0.626 \pm 0.016$ | $0.895 \pm 0.003$ | $0.578 \pm 0.002$ | $0.849 \pm 0.003$ |
| Shuffle within Trigrams | Flava | $0.613 \pm 0.007$ | $0.889 \pm 0.003$ | $0.564 \pm 0.006$ | $0.834 \pm 0.003$ |
| No Shuffling | XVLM | 0.967 | 1.000 | 0.855 | 0.968 |
| Shuffle only nouns and adj. | XVLM | $0.881 \pm 0.013$ | $0.987 \pm 0.001$ | $0.733 \pm 0.002$ | $0.920 \pm 0.002$ |
| Shuffle all words | XVLM | $0.869 \pm 0.003$ | $0.982 \pm 0.003$ | $0.708 \pm 0.001$ | $0.908 \pm 0.002$ |
| Shuffle all but nouns and adj. | XVLM | $0.937 \pm 0.003$ | $0.994 \pm 0.003$ | $0.798 \pm 0.003$ | $0.949 \pm 0.003$ |
| Shuffle trigrams | XVLM | $0.924 \pm 0.005$ | $0.995 \pm 0.002$ | $0.795 \pm 0.008$ | $0.948 \pm 0.001$ |
| Shuffle within Trigrams | XVLM | $0.942 \pm 0.013$ | $0.996 \pm 0.002$ | $0.807 \pm 0.003$ | $0.953 \pm 0.001$ |

2. BLIP (Li et al., 2022): We report results for the 'Base' variants of BLIP. For experiments with COCO, we use the version finetuned on COCO; and for experiments with Flickr30k, we use the version finetuned on Flickr. For Visual Genome experiments, we use the variant that gives better results, that is the one finetuned on COCO. We use the checkpoints released at https://github.com/salesforce/BLIP.

3. X-VLM (Zeng et al., 2022b): Similar to BLIP, we report results with the 'Base' variants of X-VLM. For experiments with COCO, we use the version finetuned on COCO; and for experiments with Flickr30k, we use the version finetuned on Flickr. We use the checkpoints released at https://github.com/zengyan-97/X-VLM/.

Table 4: Performance in the image shuffled retrieval task. Performance is averaged over 3 seeds, and standard deviations are reported next to each mean.

| Strategy | Model | Text Recall@1 | Text Recall@5 | Image Recall@1 | Image Recall@5 |
|---|---|---|---|---|---|
| COCO Image Shuffle | | | | | |
| No Shuffling | BLIP | 0.814 | 0.953 | 0.635 | 0.855 |
| Shuffle Rows (4) | BLIP | $0.692 \pm 0.005$ | $0.880 \pm 0.002$ | $0.546 \pm 0.001$ | $0.790 \pm 0.000$ |
| Shuffle Columns (4) | BLIP | $0.705 \pm 0.004$ | $0.880 \pm 0.005$ | $0.553 \pm 0.003$ | $0.795 \pm 0.001$ |
| Shuffle Patches (9) | BLIP | $0.594 \pm 0.003$ | $0.817 \pm 0.001$ | $0.488 \pm 0.002$ | $0.741 \pm 0.000$ |
| No Shuffling | CLIP | 0.503 | 0.748 | 0.301 | 0.557 |
| Shuffle Rows (4) | CLIP | $0.402 \pm 0.006$ | $0.657 \pm 0.003$ | $0.252 \pm 0.004$ | $0.488 \pm 0.002$ |
| Shuffle Columns (4) | CLIP | $0.410 \pm 0.002$ | $0.661 \pm 0.002$ | $0.256 \pm 0.001$ | $0.493 \pm 0.000$ |
| Shuffle Patches (9) | CLIP | $0.284 \pm 0.002$ | $0.531 \pm 0.012$ | $0.209 \pm 0.003$ | $0.428 \pm 0.002$ |
| No Shuffling | Flava | 0.454 | 0.788 | 0.388 | 0.682 |
| Shuffle Rows (4) | Flava | $0.305 \pm 0.007$ | $0.629 \pm 0.004$ | $0.311 \pm 0.004$ | $0.588 \pm 0.003$ |
| Shuffle Columns (4) | Flava | $0.325 \pm 0.000$ | $0.650 \pm 0.001$ | $0.318 \pm 0.000$ | $0.603 \pm 0.002$ |
| Shuffle Patches (9) | Flava | $0.182 \pm 0.004$ | $0.449 \pm 0.001$ | $0.243 \pm 0.001$ | $0.500 \pm 0.001$ |
| No Shuffling | XVLM | 0.791 | 0.947 | 0.610 | 0.848 |
| Shuffle Rows (4) | XVLM | $0.664 \pm 0.006$ | $0.876 \pm 0.005$ | $0.524 \pm 0.002$ | $0.781 \pm 0.003$ |
| Shuffle Columns (4) | XVLM | $0.687 \pm 0.003$ | $0.883 \pm 0.001$ | $0.539 \pm 0.000$ | $0.790 \pm 0.002$ |
| Shuffle Patches (9) | XVLM | $0.463 \pm 0.005$ | $0.760 \pm 0.003$ | $0.471 \pm 0.001$ | $0.731 \pm 0.001$ |
| Flickr30k Image Shuffle | | | | | |
| No Shuffling | BLIP | 0.972 | 0.999 | 0.869 | 0.974 |
| Shuffle Rows (4) | BLIP | $0.905 \pm 0.002$ | $0.980 \pm 0.005$ | $0.791 \pm 0.004$ | $0.943 \pm 0.004$ |
| Shuffle Columns (4) | BLIP | $0.918 \pm 0.007$ | $0.991 \pm 0.003$ | $0.812 \pm 0.004$ | $0.954 \pm 0.001$ |
| Shuffle Patches (9) | BLIP | $0.875 \pm 0.004$ | $0.973 \pm 0.002$ | $0.739 \pm 0.001$ | $0.920 \pm 0.003$ |
| No Shuffling | CLIP | 0.784 | 0.950 | 0.591 | 0.835 |
| Shuffle Rows (4) | CLIP | $0.687 \pm 0.020$ | $0.895 \pm 0.001$ | $0.493 \pm 0.002$ | $0.763 \pm 0.002$ |
| Shuffle Columns (4) | CLIP | $0.699 \pm 0.001$ | $0.892 \pm 0.006$ | $0.511 \pm 0.004$ | $0.769 \pm 0.005$ |
| Shuffle Patches (9) | CLIP | $0.559 \pm 0.008$ | $0.814 \pm 0.008$ | $0.411 \pm 0.011$ | $0.683 \pm 0.008$ |
| No Shuffling | Flava | 0.707 | 0.941 | 0.664 | 0.900 |
| Shuffle Rows (4) | Flava | $0.562 \pm 0.008$ | $0.849 \pm 0.008$ | $0.550 \pm 0.008$ | $0.823 \pm 0.001$ |
| Shuffle Columns (4) | Flava | $0.602 \pm 0.006$ | $0.882 \pm 0.003$ | $0.588 \pm 0.002$ | $0.852 \pm 0.004$ |
| Shuffle Patches (9) | Flava | $0.404 \pm 0.011$ | $0.725 \pm 0.006$ | $0.463 \pm 0.004$ | $0.753 \pm 0.007$ |
| No Shuffling | XVLM | 0.967 | 1.000 | 0.855 | 0.968 |
| Shuffle Rows (4) | XVLM | $0.905 \pm 0.005$ | $0.982 \pm 0.002$ | $0.775 \pm 0.001$ | $0.935 \pm 0.001$ |
| Shuffle Columns (4) | XVLM | $0.921 \pm 0.005$ | $0.990 \pm 0.003$ | $0.799 \pm 0.005$ | $0.948 \pm 0.002$ |
| Shuffle Patches (9) | XVLM | $0.855 \pm 0.007$ | $0.971 \pm 0.004$ | $0.728 \pm 0.006$ | $0.908 \pm 0.003$ |

Table 5: Performance in the Pick the Right Caption task. Performance is averaged over 3 seeds, and standard deviations are reported next to each mean.

| Dataset | Model | Accuracy |
|---|---|---|
| Flickr30k-PRC | BLIP | $0.369 \pm 0.009$ |
| Flickr30k-PRC | CLIP | $0.595 \pm 0.006$ |
| Flickr30k-PRC | Flava | $0.129 \pm 0.005$ |
| Flickr30k-PRC | XVLM | $0.473 \pm 0.003$ |
| Flickr30k-PRC | Chance-level | 0.2 |
| COCO-PRC | BLIP | $0.321 \pm 0.001$ |
| COCO-PRC | CLIP | $0.460 \pm 0.001$ |
| COCO-PRC | Flava | $0.039 \pm 0.001$ |
| COCO-PRC | XVLM | $0.362 \pm 0.003$ |
| COCO-PRC | Chance-level | 0.2 |

4. FLAVA (Singh et al., 2022): We use the model released at the Huggingface Transformers (Wolf et al., 2020) library. We follow the tutorial shared by authors, and use the "flava-full" variant[7].

---

[7] https://github.com/apsdehal/flava-tutorials/blob/main/winoground-flava-example.ipynb

Table 6: Here, we report the result of fine-tuning CLIP with hard-negatives. Metrics reported: Accuracy for CIFAR10, CIFAR100, ImageNet; Recall@1 for Flickr30k and COCO text-to-image and image-to-text retrieval; Macro Accuracy for VG-Relations and VG-Attribution; Accuracy for Flickr30k and COCO pick-the-right-caption tasks.

| | CLIP | CLIP-FT | NegCLIP |
|---|---|---|---|
| **Compositional Tasks** | | | |
| **VG-Relation** | 0.59 | 0.63 | 0.81 |
| **VG-Attribution** | 0.62 | 0.65 | 0.71 |
| **Flickr30k-PRC** | 0.59 | 0.50 | 0.91 |
| **COCO-PRC** | 0.46 | 0.36 | 0.86 |
| **Downstream Tasks** | | | |
| **CIFAR10** | 0.95 | 0.95 | 0.94 |
| **CIFAR100** | 0.80 | 0.80 | 0.79 |
| **ImageNet** | 0.75 | 0.74 | 0.72 |
| **Flickr30k Image R@1** | 0.59 | 0.67 | 0.67 |
| **Flickr30k Text R@1** | 0.78 | 0.83 | 0.79 |
| **COCO Image R@1** | 0.30 | 0.42 | 0.41 |
| **COCO Text R@1** | 0.50 | 0.59 | 0.56 |

## C  NEGATIVE MINING

### C.1  NEGATIVE TEXT MINING

Starting from the MSCOCO image dataset, we use spacy to swap the position of two elements of the caption. These elements can be either nouns, adjectives, adverbs, verb phrases and noun phrases (only if the noun phrase is composed by three tokens or more, to not overlap with noun swapping). For each caption we thus build a set of 5 possible negative captions (note that, a negative caption might have less then 5 negative captions if there are not enough elements to swap). If a caption has zero negative captions, we remove it from the dataset. During training, at each epoch for each caption we sample one of its negative captions as additional element of the batch.

### C.2  NEGATIVE IMAGE MINING

Starting from the MSCOCO image dataset, we compute the pairwise similarity between all the images. Then, for each image we collect the 3 most similar images. During training, at each epoch for each image we sample one of its negative images as additional element of the batch.

### C.3  FINE-TUNING DETAILS

For finetuning models, we build our code on https://github.com/mlfoundations/open_clip/. We finetune all models on the training split of the COCO dataset, and validate on the validation split. We finetune both CLIP-FT and NegCLIP for 5 epochs, with sweeping learning rates in $\{1e-5, 5e-6, 1e-6\}$ and picking the models based on the retrieval performance in the COCO validation set. We use 50 steps of warmup and AdamW optimizer with a cosine-annealing learning rate schedule with $N = 1024$ batch size using a single NVIDIA RTX 2080 Ti GPU.

**Limitations:** We note that we were not able to train an entire model from scratch due to computational resources. We expect future work further to verify the benefits of composition-aware negative mining in contrastive pretraining. Furthermore, note that Radford et al. (2021) use $N = 32,000$ as the batch size, while we only used a single gpu with $N = 1024$. Similarly, we expect to gain further improvements from larger batch sizes.

