# OpenReview forum: "When and Why Vision-Language Models Behave like Bags-Of-Words, and What to Do About It?"
_ICLR.cc/2023/Conference — ICLR 2023 notable top 5%_

### Official Review · Reviewer_y4fM · 2022-10-24

**Confidence:** 4
**Clarity, Quality, Novelty And Reproducibility:** The paper is of good quality and the …
**Correctness:** 4
**Technical Novelty And Significance:** 2
**Empirical Novelty And Significance:** 2
**Recommendation:** 6

**Strength And Weaknesses:**

Strengths: (1) Bring up an interesting and critical issue of not sensitive to compositional order of the current large VL models
(2) Four new tasks based on existing dataset that do not need human annotations to validating the aforementioned issue
(3) a simple yet effective treatment to the issue by adding compositional-aware hard examples
(4) Good results on the newly proposed tasks


Weakness:
(1) Hard to control each example in the 4 tasks, for example, swapping adj for the caption "a cute girl is walking a little dog", it would be good to have human check if the semantic is actually not valid


**Summary Of The Paper:**

The paper proposes four additional task for validating whether current large VL models trained with contrastive losses are sensitive to detailed attributes and relationships of the objects. The tasks include genome attributes and relations and COCO and Flickr captions. The paper found that most large VL models can not respond to the changes applied to the attributes and captions. In order to address this issue, the paper propose to add  composition-aware hard negatives during training. The resulting model obtains similar results on the original downstream task and good results on the 4 proposed new tasks.

**Summary Of The Review:**

The paper proposes 4 new tasks to measure whether a large VL model trained using contrastive loss are sensitive to compositional orders. After finding that most of them perform bad on these 4 new task, the paper propose a simple fix by adding compositional aware hard negative. These validation of 4 new tasks and the compositional hard negatives help the model be more compositional aware without much suffering on the original  downstream tasks. My largest concern is that when using the 4 tasks as validation dataset, the dataset should eb further validated by human to check if the semantic is valid or not.

---

> ### Author Response · Authors · 2022-11-11
> **Author Response to Reviewer y4fM**
>
> Dear Reviewer y4fM,
>
> Thank you for your detailed review! You evidently spent a substantial time reviewing our paper, and we are thankful for your efforts during the challenging 2 week review period. We appreciate your comments that we “bring up an interesting and critical issue” and that we have “a simple yet effective treatment”.
>
> **Semantic Validity:** First, we note how we get the attributes for the Visual Genome Attribution dataset. We mine the attributes “that are not mutual” between two objects. Namely, if “small” is a common attribute between the two objects, we do not include it. Hence, when the attributes in the test cases are swapped, it is not very likely to include common attributes. We argue that the probability of the caption being semantically valid after swapping is very low.
>
> **Validation study:** We took your suggestion to use human validation and ran the following verification study:
> To estimate the error probabilities, we randomly sampled 100 instances from the VG-Attribution dataset and 100 instances from the VG-Relation dataset. Next, we recruited 2 reviewers, who were asked to pick between 4 options for each image: Caption 1, Caption 2, None, or Both. The order of captions is randomly swapped.
>
> Results:
> VG Attributions: 99/100 samples correct caption; 1/100 samples false caption, 0/100 labeled as None/Both.
> VG Relations: 93/100 samples correct caption; 4/100 samples false caption, 3/100 labeled as None 0/100 labeled as both.
>
> Overall, we believe these demonstrate that the semantic validity of a permuted caption for the given image is a low-probability event. We are posting a subset of our dataset at https://anonymous.4open.science/r/aro-benchmark-samples-65E0/  for you to investigate if desired. The full dataset will be released upon deanonymization.
>
>
> Thank you very much for your thoughtful review! We believe your comments led us to demonstrate the validity of our tasks. Please let us know if you have further questions, we are happy to follow up.

---

### Official Review · Reviewer_LKvk · 2022-10-24

**Confidence:** 4
**Clarity, Quality, Novelty And Reproducibility:** See above.
**Correctness:** 3
**Technical Novelty And Significance:** 3
**Empirical Novelty And Significance:** 2
**Recommendation:** 6

**Strength And Weaknesses:**

Strength:
1) This paper is well-written and easy to follow.
2) The discovery that existing visual-language models could not handle complex relationship understanding seems to be well-supported.

Weakness:
1) The proposed benchmark seems to be very similar as the Winoground, which is also cited in the related work. This, to some extent, limits the novelty of this work to the community. Could authors test the proposed composition-aware hard negatives on the Winoground to see if the method can improve these natural images capturing object relationship?

2) The proposed composition-aware hard negatives needs to know the targeted composition beforehand, which greatly limits the generalization and the potential effectiveness of this work.

3) "Understanding and Improving Robustness of Vision Transformers through Patch-based Negative Augmentation" by Qin et al is closely related to this work and should be cited in Related work.

**Summary Of The Paper:**

This paper proposed a benchmark for evaluating the ability of vision-language models with regard to relations, attributions and order information. By testing existing visual-language models, they show that the exiting models have poor relationship understanding. They further propose a simple modification contrastive loss to help models become more sensitive to orders and compositionality.

**Summary Of The Review:**

This work investigates the ability of existing visual-language models on relationship understanding and constructed a new complementary benchmark for this task. The proposed composition-aware hard negatives have major limitations but good to see they have effects on the targeted compositionality. I am leaning towards accepting this work and curious about the results on Winoground.

---

> ### Author Response · Authors · 2022-11-11
> **Author Response to Reviewer LKvk**
>
> Dear Reviewer LKvK,
>
> Thank you for your detailed review! You evidently spent a substantial time reviewing our paper, and we are thankful for your efforts during the challenging 2 week review period. We appreciate your comments saying “the paper is well-written and easy to follow”, and finding our claims well-supported.
>
> **Comparison to Winoground**: We agree that Winoground is a wonderful first step to study this problem. We first refer to a recent paper “Why is Winoground Hard? Investigating Failures in Visuolinguistic Compositionality” (Diwan et al. EMNLP, 2022). Fundamentally, Winoground intertwines several challenges beyond compositionality such as ambiguous images and tests requiring common-sense reasoning and world knowledge. As quick examples, see at https://huggingface.co/datasets/facebook/winoground/viewer/default/test sample id 44 ("the heavy oncoming traffic is contrasted with the light outgoing traffic") / 70 ("the person with the lighter shoes is holding an emoji and the person with the darker shoes isn't"). In our work, we aimed to isolate this effect and focus on testing for compositionality, without overloading with other kinds of knowledge. Hence, we not only scale the number of samples but also propose a much more specific and targeted testbed. Overall, Winoground is a great test to evaluate broad Visio-linguistic understanding; but due to its small size (400 examples) and the set of non-compositional difficulties, our dataset offers a complementary evaluation.
>
> We tested NegCLIP on Winoground and we find that it marginally improves the performance (30.5% = 30.5% overall score, 11.5% > 10.5% image score, 8% = 8% text score where Overall Score is matching all of the pairs correctly, Image Score is retrieving the correct caption for each image; see the Winoground paper for details) over CLIP. In fine-grained subsets, we observe both improvements (Relation Tag: Text 26% > 22% (N=233), Group 7% > 6%; Series Tag Text 26% > 9% (N=23)) and some performance losses. Due to the smaller sample size of Winoground, it is difficult to say if the changes are statistically significant.
>
>
> **Limitation of targeted compositions**: We agree that this approach could be further improved. We’d like to propose the vision analogues. We want vision classifiers to be invariant to rotations or flips, hence many models have them as data augmentations during training. Our approach is very similar in that we are encoding our inductive biases in augmentations. Yet, we definitely agree that further research could improve this, e.g. with Language Model Priors! In fact, to back this up, we refer to the latest developments in text-to-image models. Imagen performs better at the compositional tasks than DALL-E (Saharia et al. 2022). Even though the methodologies are mostly similar, we believe the main difference is that Imagen is using T-5 as the text encoder, versus DALL-E which uses CLIP! We believe this is related to our findings, and as you also suggested, there should be an interesting avenue for research in getting the best of both worlds, we added this as a comment to our paper. We believe our paper makes a case for further research in this venue.
>
> **Citation**: Thank you for pointing us to this paper! We added the corresponding citation both in related works and patch permutations.
>
> Thank you very much for your thoughtful review! We believe your comments led us to have a more complete paper. Please let us know if you have further questions, we are happy to follow up.

---

### Official Review · Reviewer_U4Mo · 2022-10-24

**Confidence:** 4
**Correctness:** 3
**Technical Novelty And Significance:** 3
**Empirical Novelty And Significance:** 3
**Recommendation:** 8

**Clarity, Quality, Novelty And Reproducibility:**

Quality:
- Experimental setup is thorough. My main concern would be that this becomes a new benchmark to achieve such that people interpret success on it to imply good compositional understanding. I think it works as a "bare minimum" evaluation, but performing well on it doesn't necessarily confirm a model is performing correct compositional reasoning (partially because I imagine using a LLM prior will downweight the grammatically incorrect or semantically implausible perturbations, for example).

Clarity:
- Paper is very clearly written, with a few small comments on readability of figures.

Originality:
- Relatively original; the experiments are new as far as I can tell, as is the proposed training method. However, in terms of a dataset, there are many that evaluate compositionality with synthetically generated data (GQA, CLEVR), or could be adapted to evaluate more semantically plausible alternatives for compositionality. Missing citation for the NLVR(2) corpora (Suhr et al. 2017/2019) which also evaluate compositionality with both true and false image pairs.

**Strength And Weaknesses:**

Strengths:
- The discussion of the problem and presentation of the dataset and solution is extremely clear; the paper is very well-written.
- The paper presents a clear problem with existing systems and the drawback of using retrieval as an evaluation method. The presented benchmark is offering a "bare minimum" sort of evaluation for such systems.

Weaknesses:
- The fonts in some of the figures is really small. I also strongly suggest putting numbers on the actual bar charts; it's difficult to interpret the results otherwise.
- The numbers for the experiment in 2.3 (evaluating COCO/Flickr30k on perturbations of captions) should be in the main paper, not the appendix.
- It is suggested that the reason these models ignore word ordering so much is that they really are trained as keyword identifiers, as required for image retrieval, and there's no reason to learn ordering. However, what happens if you incorporate better priors on the captions? I would imagine that a large language model would place very low probability on most of the perturbed captions (I could be wrong, though), and a VLM that uses features from a general large language model would be able to distinguish the obviously grammatically incorrect examples from the true caption.
- I was hoping the dataset would be a bit more of a scaled-up Winoground dataset, because Winoground directly tests all four settings (perturbation of relations in text and perturbation of relations in image). However, this evaluation set only seems to test perturbation of relations in text.

Questions:
- Why are there so few relations and attribute pairs in ARO, as described in Section 2.1?
- Practically, how are some of the perturbations done? With operations on top of the parse tree?
- Does the experiment in Section 4 backprop through both text and image features in CLIP?

**Summary Of The Paper:**

This paper probes the compositional generalization ability of large pre-trained vision-and-language models, such as CLIP. They introduce a benchmark dataset of synthetically generated caption perturbations, called ARO. They show that out-of-the-box performance on ARO is poor, and that existing models do not seem to distinguish captions with ordering perturbation. They also propose a method for training these models to be more robust compositionally, by mining and generating hard negative examples to fine-tune with.

**Summary Of The Review:**

This paper introduces a new benchmark and training method for evaluating compositionality of VLMs. The compositionality is mostly evaluated by perturbing captions of existing datasets in several ways. Experiments find that SOTA VLMs perform poorly (i.e., they cannot easily distinguish between perturbed and true captions). The paper is very well written. My main concern is that the kinds of perturbations in the evaluation set may be easy to reject simply by considering priors from an LLM, something which is not evaluated in this paper.

---

> ### Author Response · Authors · 2022-11-11
> **Author Response to Reviewer U4Mo**
>
> Dear Reviewer U4Mo,
>
> Thank you for your detailed review! You evidently spent a substantial time reviewing our paper, and we are thankful for your efforts during the challenging 2 week review period. We appreciate your kind comments, finding our “ solution is extremely clear”; and “the paper is very well-written”.
>
> **Format and citations**: We updated the paper to include the numbers from the tables in Figure 1-2 to have the results. Please see the updated figures that include concrete numbers in addition to the plots. We also added the citations we missed, thank you for pointing them out.
>
> **Having language priors**: We are very excited about this comment. We imagine that there will be a fruitful research avenue on merging the power of language models with multimodal training.  In fact, to back this up, we refer to the latest developments in text-to-image models. Imagen performs better at the compositional tasks at DALL-E (Saharia et al. 2022). Even though the methodologies are mostly similar, we believe the main difference is that Imagen is using T-5 as the text encoder, versus DALL-E which uses CLIP! We believe this is tightly related to our findings, and as you also suggested, there should be an interesting avenue for research in getting the best of both worlds, we added this as a comment to our paper. We believe our paper makes a case for further research in this venue.
>
> **Comparison to Winoground**: We agree that Winoground is a wonderful first step to study this problem. We first refer to a recent paper “Why is Winoground Hard? Investigating Failures in Visuolinguistic Compositionality” (Diwan et al. EMNLP, 2022). Fundamentally, Winoground has further challenges beyond compositionality such as ambiguous images and tests requiring common-sense reasoning and world knowledge. As quick examples, see at https://huggingface.co/datasets/facebook/winoground/viewer/default/test sample id 44 ("the heavy oncoming traffic is contrasted with the light outgoing traffic") / 70 ("the person with the lighter shoes is holding an emoji and the person with the darker shoes isn't") or more. Here, we aimed to isolate this effect and focus on testing for compositionality, without overloading with other kinds of knowledge. Hence, we not only scale the number of samples but also propose a much more specific and targeted testbed. Overall, Winoground is a great test to evaluate broad Visio-linguistic understanding; but due to its size (400 examples) and the set of non-compositional difficulties, our dataset offers a complementary evaluation.
>
> **Number of relations / attribute pairs**: In Appendix A.1, we describe the dataset generation procedure. To keep the annotation quality high, we filter the objects that are small in the image and hence may be hard to see (i.e. we require both height and width to be at least 1/4th of the image), and this eliminates a large set of attributes/relations from Visual Genome.
>
> **How perturbations are done**: We use Spacy to do part-of-speech tagging, and then perform the perturbations using POS tags. We will release the code that we used to generate the perturbations.
>
> **Backprop through both text and image features in CLIP?**: Yes, we backprop through both text and images.
>
> **Tracking the progress / "bare minimum" evaluation**: We agree with this sentiment. For instance, we do not expect a model to perform well in Winoground without performing well on our benchmark, as ours is a more isolated and focused evaluation. It was also indeed an intriguing surprise that the models are even not doing well with this bare minimum (grass is eating the horse vs horse is eating the grass)! Hence, we believe our evaluations are important to track the progress. On the other hand, we believe our methodology to generate test cases from Visual Genome can be used to implement more complex tasks, which can be helpful for future work.
>
> Overall, we enjoyed reading your comments and discussion, and hope that our comments clarify your questions. Please let us know if you have further questions or comments, and thank you for being very thorough with your review!

---

### Official Review · Reviewer_J2a5 · 2022-10-26

**Confidence:** 4
**Clarity, Quality, Novelty And Reproducibility:** The paper is clear and easy to follow.
**Correctness:** 3
**Technical Novelty And Significance:** 3
**Empirical Novelty And Significance:** 3
**Recommendation:** 8

**Strength And Weaknesses:**

There is no major weakness. I will just note two issues I noticed which do not affect my rating of the paper.

1.  In terms of the three challenges presented in the paper, it seems they can be addressed by explicitly considering how the challenge is formed (e.g., adding shuffled captions during training as hard negatives). However, this does not imply that the problem is solved. This reminds me of what is discussed in Teney et al., 2020, where once we know how the “challenge” dataset is constructed, we could use such information to train a model that performs well on the challenge dataset; but the model will likely fail on cases not foreseen during dataset creation.

   I wonder what should be the intended use of the benchmark. If simply adding shuffled captions can solve the challenges pretty well, what should be the next move?

Teney et al., On the Value of Out-of-Distribution Testing: An Example of Goodhart's Law.

2. I would have appreciated more discussions on relation to negative mining and contrastive learning.

---

Minor comments:

1. Most tables are in the appendix. While figures send a strong message, I would appreciate having a few tables in the main paper.

2. NegCLIP has two improvements: generating neg captions and sampling hard-neg images. Is there an ablation study on this?

3. What’s the batch size of fine-tuning CLIP?

**Summary Of The Paper:**

The paper reveals an interesting and important failure pattern of the pre-trained vision-language models (VLMs): they are insensitive to object attributes, relations, and even word orders. They created several tests from Visual Genome, COCO caption, and Flickr30K, and show several representative VLMs (BLIP, CLIP, etc.) are insensitive to:

1. Relation (prepositions and verbs): e.g., differentiate between “the **man** is behind the **tree**’ vs “the **tree** is behind the **man**”

2. Attribute: e.g., differentiate between “the **crouched man** and the **open door**” and “the **open man** and the **crouched door**”

3. Word order: whether the model can tell if a sentence differentiates a sentence with shuffled words

The paper goes on discussing what causes the models to be insensitive to such compositional structures in the images and captions. They provide an intuitive hypothesis: the contrastive (retrieval) objective in the pre-training does not encourage the model to learn such compositional structures, because “the datasets are not designed to contain many images with captions containing similar words that must be differentiated” (or from my understanding, the probability of being able to sampling such “hard negatives” is very small). They also provided an experiment backing it up: models do not need order information to do well on current image-caption retrieval benchmarks.

They further show that if we purposely generate negative captions by swapping the word orders and sample hard negative images using clip similarity, the model does much better on the proposed evaluation benchmarks.

**Summary Of The Review:**

Overall, this is a well-presented paper with an easy-to-follow yet important finding: performance on the current VL benchmarks could be misleading (even though these benchmark data contain images and captions of rich compositional structures) and current VLMs are very close to bag-of-word models.

---

> ### Author Response · Authors · 2022-11-11
> **Author Response to Reviewer J2a5**
>
> Dear Reviewer J2a5,
>
> Thank you for your detailed review! You evidently spent a substantial time reviewing our paper, and we are thankful for your efforts during the challenging 2 week review period. We are very excited about your approval, and kind comments saying our work has “no major weakness”, and has “easy-to-follow yet important findings”.
>
> **Tracking the progress**: We share your sentiment that performing well on our tasks will not be the end goal, and we view our benchmark as a first, and isolated step. To motivate this, we refer to the concurrent work where Winoground has been suggested to include broader challenges, such as Text Difficulties / Complex Reasoning and more (“Why is Winoground Hard? Investigating Failures in Visuolinguistic Compositionality”. Diwan et al. EMNLP, 2022). In this sense, we believe our work isolates these effects and focuses on tracking the progress focusing on compositional sensitivity.
>
> **What’s next**: We also note that there is still further improvement to expect, both in our challenges and definitely on Winoground! Broadly speaking, hard negative mining is an augmentation strategy to encode our inductive biases, but certainly does not mean inheriting all the desired properties of language. We believe there is a very exciting area of future work that aims to inherit (and evaluate!) the language model priors to improve the understanding of VLMs. We hope that our work motivates this branch.
>
> **Minor comments**: We updated the paper to include the numbers from the tables in Figure 1. Please see the updated figures that include concrete numbers in addition to plots.  We are sorry to miss the batch size in the text, it is $N=1024$ and we updated the paper to reflect this. In preliminary experiments, we observed that having both neg captions and neg images presents the best results. We will add the results of the ablation to the camera-ready version.
>
> We again thank you for your detailed comments, exciting intellectual query on what is the bigger picture, and for helping us make our paper better. Please let us know if you have further questions or comments, we are happy to follow up!

---

### Public Comment · ~Zehao_Wang6 · 2023-07-09
**I cannot reproduce Table 6**

Hello, thank you for your excellent and creative work. I would like to know whether the performance of Zero Shot classification is reported in Table 6 of the paper. Why did I test the [checkpoint](https://github.com/mertyg/vision-language-models-are-bows/blob/main/model_zoo/__init__.py#L66) of your released model with [clip-benchmark](https://github.com/LAION-AI/CLIP_benchmark/tree/main) and get different results?

I doubt that Table 6 reports the accuracy of Finetune on downstream data sets, is that right？

This is my command: `clip_benchmark eval --dataset=imagenet --task=zeroshot_classification  --pretrained=./negCLIP.pt --model=ViT-B-32 --output=result.json --batch_size=64
`

This is the result of comparison:

|          | Your paper | My test |
|----------|------------|---------|
| CIFAR10  |   94.0     |  85.9   |
| CIFAR100 |   79.0     |  60.9   |
| ImageNet |   72.0     |  55.7   |

Look forward to your reply. Thank you very much

---

> ### Author Response · Authors · 2023-07-09
> **Linear probing**
>
> Hello!
>
> Results in Table 6 show linear probing performance! See [here](https://github.com/mertyg/vision-language-models-are-bows/issues/29).

---

### Decision · Program_Chairs · 2023-01-20

**Decision:**

Accept: notable-top-5%

**Justification For Why Not Higher Score:**

N/A

**Justification For Why Not Lower Score:**

The problem studied in this paper is very relevant for visual representation learning. The authors have done a good job of finding failure modes for most vision-language models and proposing a sensible benchmark to evaluate it. I believe this deserves attention from the ML community at large given the increased interest in this field.

**Metareview: Summary, Strengths And Weaknesses:**

*Summary*: This paper studies vision-language models that are trained using a contrastive alignment objective for the image and language representations. The authors find interesting and practically important failure modes of such models - insensitivity to word order, Subject-object relations, and other compositional failures. They propose a benchmark to evaluate these models since current benchmarks do not reveal these failures. Finally, they propose a hard negative mining approach that improves results.

*Strengths*: (1) The central finding of failure modes for vision-language models is stated and evaluated clearly. I believe this paper was written really well and easy to follow. (2) The authors propose a benchmark to evaluate the failure modes which is useful for future work. (3) The authors propose an automatic way to create compositional hard negatives that improve performance.

*Weaknesses*: (1) The benchmark proposed in this work, as noted by other reviewers, is not extensive as it only tests for perturbations in the text. (2) The automatic hard negative mining process  relies on extra information which limits its broad applicability.

**Note From Pc:**

if the above contains the word "oral" or "spotlight" please see: "oral" presentation means -> notable-top-5% and "spotlight" means -> notable-top-25%. As stated in our emails, we are disassociating presentation type from AC recommendations